# STRUCTURE BEFORE THE MACHINE: INPUT SPACE IS THE PREREQUISITE FOR CONCEPTS

## ABSTRACT

High-level representations have become a central focus in enhancing AI transparency and control, shifting attention from individual neurons or circuits to structured semantic directions that align with human-interpretable concepts. Motivated by the Linear Representation Hypothesis (LRH), we propose the Input-Space Linearity Hypothesis (ISLH), which posits that concept-aligned directions originate in the input space and are selectively amplified with increasing depth. We then introduce the Spectral Principal Path (SPP) framework, which formalizes how deep networks progressively distill linear representations along a small set of dominant spectral directions. Building on this framework, we further demonstrate the multimodal robustness of these representations in Vision-Language Models (VLMs). By bridging theoretical insights with empirical validation, this work advances a structured theory of representation formation in deep networks, paving the way for improving AI robustness, fairness, and transparency.

## 1 INTRODUCTION

Deep learning has achieved remarkable success across various domains, including computer vision (Krizhevsky et al., 2012), natural language processing (Devlin et al., 2019), and speech recognition (Hinton et al., 2012; Graves et al., 2013). However, the internal mechanisms of neural networks remain opaque. Despite advances in visualization and interpretability techniques, the transformation of inputs into high-level representations and the interactions among neurons are still not fully understood (Lipton, 2016; Doshi-Velez & Kim, 2017b; Ribeiro et al., 2016). This lack of transparency leads to the characterization of neural networks as "black boxes" (Lipton, 2016; Doshi-Velez & Kim, 2017b), raising concerns about their reliability, particularly in high-stakes applications such as healthcare (Caruana et al., 2015), finance (Rudin, 2019), and law (Doshi-Velez & Kim, 2017a).

Previous works have demonstrated the potential of representations as a new perspective on AI transparency. For example, neural networks trained to play chess exhibit internal representations of board positions and strategies (McGrath et al., 2022). Similarly, both generative and self-supervised models have been shown to develop emergent representations, such as semantic segmentation in vision tasks (Caron et al., 2021; Oquab et al., 2023). Zou et al. (2023) further formalized Representation Engineering (RepE), emphasizing its ability to extract meaningful concepts from a model's internal structure and control model behavior. RepE has emerged as a top-down approach to enhance the model transparency that focuses on representations rather than individual neurons or circuits, providing a more structured understanding of AI transparency and control. Another important contribution is the Linear Representation Hypothesis (LRH) (Park et al., 2023): as depth increases, task-relevant concepts become nearly linearly separable in the model's latent space, making them accessible with simple probes or linear edits.

Despite these promising advances, existing works on representations remain largely observational, relying on observed phenomena or intuitions. RepE uses contrastive pairs (e.g., honesty vs. dishonesty) to surface concept directions, but further theoretical work is needed to clarify why and how such directions emerge and remain coherent across layers. Similarly, LRH assumes linearity in embedding and unembedding spaces, yet offers limited insight into why representations become linearly organized. These approaches typically do not address how representations scale or propagate through deep networks, leaving a gap in our understanding of their robustness, generality, and theoretical foundations.

In this work, we move beyond linear observations by introducing Spectral Principal Path (SPP) that explains the emergence and stability of linear representations in deep networks. We show that representations propagate through a small number of spectral principal paths—directions aligned with large singular values at each layer. This structure naturally explains why concept directions remain stable and linearly accessible across layers, offering a theoretical foundation for both RepE and the Linear Representation Hypothesis. We further extend this analysis to Vision-Language Models (VLMs), demonstrating how spectral dynamics govern the interaction between visual and linguistic modalities. Our framework not only bridges theory and practice but also provides concrete tools to improve robustness and interpretability in multimodal AI systems.

Our main contributions are as follows:

- **Input-Space Linearity Hypothesis (ISLH).** We extend the Linear Representation Hypothesis beyond embedding and unembedding spaces to the input space itself, showing that concept directions can be traced backward to the input space.
- **Spectral Principal Path (SPP)**. We propose a principled mechanism explaining how representations propagate and stabilize across layers via a small number of spectral principal paths—directions aligned with large singular vectors.
- **Multimodal robustness of representations**. We evaluate Representation Engineering in VLMs and demonstrate that linearly organized concept representations remain robust across modalities. This provides the first empirical validation of RepE's scalability in multimodal systems and supports the generality of spectral structure.

## 2 RELATED WORKS

### 2.1 REPRESENTATIONS IN NEURAL NETWORKS

Early work on word embeddings shows that neural networks can learn distributed representations that encode semantic relationships and compositional structures (Mikolov et al., 2013). Follow-up studies (Schramowski et al., 2019; Radford et al., 2015) further reveal that learned embeddings can implicitly encode abstract dimensions such as commonsense morality, even without explicit supervision. For instance, Radford et al. (Radford et al., 2015) observe that training a language model on product reviews results in the emergence of a sentiment-tracking neuron.

This phenomenon is not unique to language models. McGrath et al. (2022) show that similar internal representations can be found in networks trained to play chess. In computer vision, recent studies (Caron et al., 2021; Oquab et al., 2023) demonstrate that both generative and self-supervised training objectives give rise to emergent semantic representations, such as those useful for segmentation tasks, suggesting the emergence of representations is a general property of deep learning systems.

Building on this, Zou et al. (2023) propose techniques to read and control these internal structures, including Linear Artificial Tomography (LAT) for extracting concept-aligned representations and methods for steering model behavior. Their study shows that RepE-style approaches can be used not only to detect but also to manipulate emergent properties, motivating more systematic efforts to characterize and intervene in high-level model behaviors. Theoretically, Park et al. (2023) proposes the Linear Representation Hypothesis: task-relevant concepts become nearly linearly separable in the model's latent space.

### 2.2 APPROACHES TO INTERPRETABILITY

Traditional interpretability techniques have focused on methods like saliency maps (Simonyan et al., 2013; Springenberg et al., 2014; Zeiler & Fergus, 2014; Zhou et al., 2016), feature visualization (Szegedy et al., 2013; Zeiler & Fergus, 2014) and mechanistic interpretability (Olah et al., 2020; Olsson et al., 2022; Lieberum et al., 2023). Saliency maps (Simonyan et al., 2013) highlight important input regions by tracking gradients or activation values, yet they are often unstable and provide limited insight into the distributed nature of representations. Similarly, feature visualizations (Szegedy et al., 2013; Zeiler & Fergus, 2014) optimize inputs to activate specific neurons, but they may overlook the global structure of the emergent representations. Mechanistic interpretability (Zou et al., 2023) seeks to fully reverse engineer neural networks into their "source code", but the

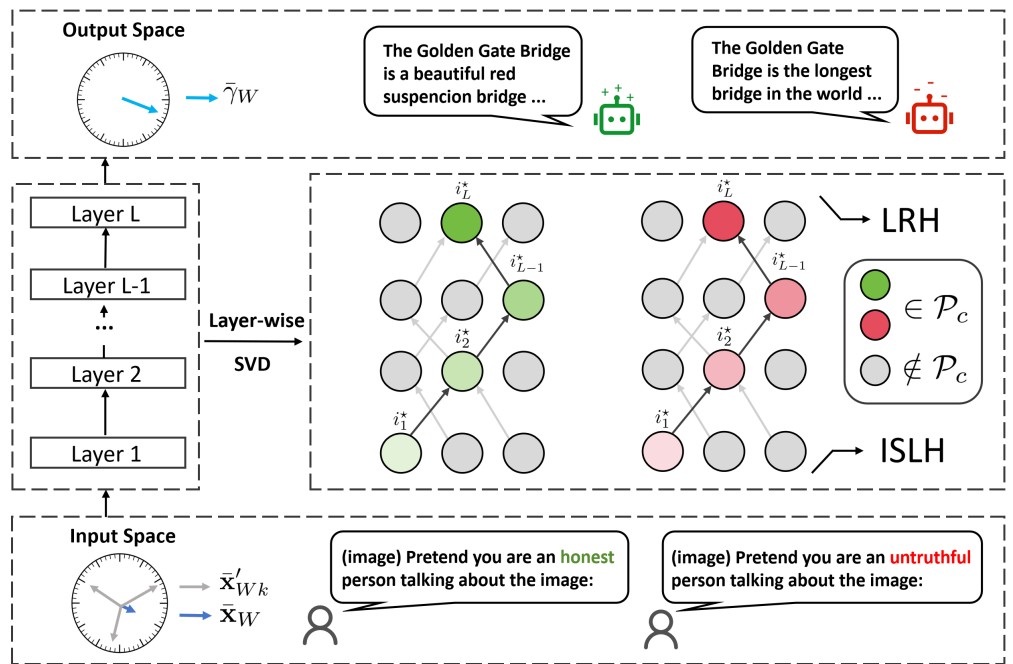

Figure 1: Overview of the Spectral Principal Path framework: the illustration shows how deep networks extract and amplify concept-relevant directions from the input, culminating the linear representation hypothesis. $\bar{x}_W$ is the concept discriminative direction in the input space, $\bar{\mathbf{x}}'_{Wk}$ are spurious directions, $\bar{\gamma}_W$ is the direction vector in the unembedding space, and the spectral path $\{i_1^*, \ldots, i_L^*\}$ is constructed by selecting the top singular direction at each layer via SVD on weight matrices.

considerable manual effort and the difficulty of theoretically explaining neural networks as discrete circuits hinder their explainability.

In contrast, recent advances in interpretability have shifted the focus toward analyzing representation spaces. This top-down approach seeks to uncover high-level semantic directions that correspond to complex phenomena such as honesty, fairness, or bias. By extracting and analyzing these internal representations, researchers have opened new avenues to understand how large-scale AI models encode and preserve crucial information across layers, leading to more robust and interpretable AI systems.

## 3 PRELIMINARIES

**Linear Representation Hypothesis** (Park et al., 2023) We consider a co ncept $W$ that has a *linear representation* in a model if there exists a vector $\bar{\gamma}_W$ in the unembedding space $\Gamma$ and a vector $\bar{\lambda}_W$ in the embedding space $\Lambda$ such that for any counterfactual pair $(Y(W = 0), Y(W = 1))$,

$$\gamma(Y(W = 1)) - \gamma(Y(W = 0)) \in \text{Cone}(\bar{\gamma}_W), \tag{1}$$

and for any context pair $(\lambda_0, \lambda_1)$ that changes only $W$ and not other causally separable concepts,

$$\lambda_1 - \lambda_0 \in \text{Cone}(\bar{\lambda}_W). \tag{2}$$

where $\text{Cone}(\mathbf{v}) = \{\alpha \mathbf{v} : \alpha > 0\}$, the embedding space is where input contexts are mapped to high-dimensional vectors before processing, capturing the model's internal representation of the input. The unembedding space is where each output token is represented, and predictions are made by computing inner products between input embeddings and output unembedding vectors. Unless stated otherwise, all discussions pertain to the embedding space $\Lambda$, as our goal is to trace how input linearity propagates through the network.

# 4 SPECTRAL PRINCIPAL PATH FRAMEWORK

The overview of the Spectral Principal Path framework is shown in Fig. 1. In the input space, the concept direction $\bar{\mathbf{x}}_W$ separates inputs with contrast concepts such as "honest" and "untruthful". As activations propagate through the network, layer-wise SVD identifies spectral components, forming spectral principal paths $\mathcal{P}_c$. These dominant paths progressively amplify concept-relevant signals, leading to output representations linearly aligned with $\bar{\gamma}_W$.

## 4.1 INPUT-SPACE LINEARITY HYPOTHESIS

Inspired by LRH, which uncovers linear concept axes in embedding and unembedding spaces, we take one step further and ask whether such axes already reside in the *raw input space*. Input-Space Linearity Hypothesis assumes that, in the *raw input space* $x \in \Upsilon$, there exists a discriminative direction $\bar{\mathbf{x}}_W$ such that

$$\mathbb{E}[x \mid W=1] - \mathbb{E}[x \mid W=0] \in \mathrm{Cone}(\bar{\mathbf{x}}_W), \tag{3}$$

yet each sample is an entangled mixture

$$x^{(i)} = \alpha_i \bar{\mathbf{x}}_W + \sum_{k=1}^{r} \beta_{i,k} \bar{\mathbf{x}}'_{Wk} + \varepsilon_i, \tag{4}$$

where $\bar{\mathbf{x}}'_{Wk}$ are spurious directions and $\varepsilon_i$ is residual noise. ISLH states that for any intervention flipping only $W$, the induced input difference satisfies $\mathrm{Cone}(\bar{\mathbf{x}}_W)$.

ISLH pinpoints the origin of linearity by showing that concept axes already reside in raw input coordinates and are merely recovered and amplified during training; where training can be viewed as a noise-suppression process, where spectral principal paths with large singular values progressively dampen spurious components $\beta_{i,k}\bar{\mathbf{x}}'_{Wk}$; and, by grounding linearity at the input level, it becomes inherently modality-agnostic, extending Representation Engineering to multimodal models whose raw signals already encode task-relevant contrasts. Next, we will dive into the connection between ISLH and LRH:

**Theorem 4.1** (ISLH sufficiency). *If the network satisfies the* Input-Space Linearity Hypothesis *(ISLH), and the representation dominates the cumulative gain $G(\mathcal{P})$ (shown in (12)), then its deep representations satisfy the* Linear Representation Hypothesis *(LRH); that is, concept classes become linearly separable in the latent space.*

The proof is given in Appendix A.1.

## 4.2 SPECTRAL PRINCIPAL PATH

We are now asking how such concept directions propagate through the network. While ISLH posits that concept-aligned directions already exist in the raw input space, it does not yet explain why these directions persist and become more prominent across layers. To address this, we introduce the *Spectral Principal Path* (SPP) framework, which shows that representations are distilled through a small set of principal spectral paths aligned with large singular vectors at each layer. This framework formalizes how ISLH leads to the emergence of the Linear Representation Hypothesis (LRH), providing a unified and mechanistically grounded view of representation stability.

Specifically, consider a generalized network

$$f_L(x) = W_L W_{L-1} \cdots W_1 x \equiv Mx, \qquad W_l \in \mathbb{R}^{d_l \times d_{l-1}}, \tag{5}$$

according to LRH, there exists a representation direction $\bar{\lambda}_W$, where the neural activity $f(x)$ can be linearly projected into that direction, formulating a representation score:

$$s(x) = \langle \bar{\lambda}_W, \ f_L(x) \rangle = \bar{\lambda}_W^\top Mx, \qquad \bar{\lambda}_W \in \mathbb{R}^{d_L}. \tag{6}$$

While our theoretical formulation assumes a purely stacked linear architecture, we show our extension to residual connections and attention mechanisms. We provide a detailed discussion of these extensions in Appendix A.2.1.

Next we will calculate the back-propagated gradient of $s$ using the chain rule,

$$\nabla_x s = \left(\prod_{l=1}^{L} \nabla f_{l \to (l-1)}\right)^{\top} \bar{\lambda}_W, \tag{7}$$

$$\nabla f_{l \to (l-1)} = W_l + \sum_k f_{l-1,k} \frac{\partial W_l}{\partial f_{l-1,k}}. \tag{8}$$

To make the structure of the layer-wise Jacobian in (8) explicit, we regard the gradient $\nabla f_{l \to (l-1)}$ as a matrix and apply its compact singular-value decomposition (SVD); this yields

$$\nabla f_{l \to (l-1)} = U^{(l)} \Sigma^{(l)} V^{(l)\top}, \quad \Sigma^{(l)} = \mathrm{diag}\big(\sigma_1^{(l)}, \dots, \sigma_{r_l}^{(l)}\big), \tag{9}$$

therefore

$$\nabla_x s = V^{(1)} \Sigma^{(1)} U^{(1)\top} \cdots V^{(L)} \Sigma^{(L)} U^{(L)\top} \bar{\lambda}_W. \tag{10}$$

Unfolding the matrix products yields

$$\nabla_x s = \sum_{i_1,\dots,i_L} \left(\prod_{l=1}^{L} \sigma_{i_l}^{(l)}\right) V_{\cdot i_1}^{(1)} \left(\prod_{l=1}^{L-1} \langle u_{i_l}^{(l)}, V_{\cdot i_{l+1}}^{(l+1)} \rangle \right) \langle u_{i_L}^{(L)}, \bar{\lambda}_W \rangle , \tag{11}$$

where $\sigma_{i_l}^{(l)}$ is the signular value within the $\Sigma^{(l)}$ matrix, $u_{i_l}^{(l)}$ (resp. $V_{\cdot i_l}^{(l)}$) is the $i_l$-th left (resp. right) singular vector, and $\cdot i_l$ here means select the $i_l$-th column. Therefore equation 11 is dominated by paths whose cumulative gain is largest. Formally, we define Spectral Principal Path as follows:

**Definition 4.1** (Spectral Principal Path). Given the Jacobian decomposition across $L$ layers $\nabla_x s$, each spectral path $\mathcal{P} = (i_1, \dots, i_L)$ contributes a cumulative gain given by

$$G(\mathcal{P}) := \left(\prod_{l=1}^{L} \sigma_{i_l}^{(l)}\right) V_{\cdot i_1}^{(1)} \left(\prod_{l=1}^{L-1} \langle u_{i_l}^{(l)}, V_{\cdot i_{l+1}}^{(l+1)} \rangle \right) \langle u_{i_L}^{(L)}, \bar{\lambda}_W \rangle , \tag{12}$$

the **Spectral Principal Path** is defined as $\mathcal{P}_c = (i_1^\star, \dots, i_L^\star)$ that maximizes the cumulative gain:

$$\mathcal{P}_c = (i_1^\star, \dots, i_L^\star) := \arg \max_{(i_1,\dots,i_L)} G(\mathcal{P}). \tag{13}$$

### 4.3 Connection between ISLH and SPP

To clarify how information specified by ISLH is propagated along an SPP, we introduce the notion of *spectral similarity* for a given spectral path $(i_1, \dots, i_L)$:

**Definition 4.2** (Spectral Similarity). For two consecutive layers $l$ and $l+1$ in the unfolded Jacobian, and for indices $(i_l, i_{l+1})$, we define the *spectral similarity at layer $l$* as

$$\Theta(i_l, i_{l+1}) := \big\langle u_{i_l}^{(l)}, V_{\cdot i_{l+1}}^{(l+1)} \big\rangle, \qquad l = 1, \dots, L-1. \tag{14}$$

This quantity measures how well the $i_l$-th spectral component of layer $l$ aligns with the $i_{l+1}$-th spectral component that enters layer $l+1$.

Empirically, we observe two coupled effects as the network reaches deeper.

1. **Stabilization of singular vectors.** As demonstrated in Fig. 2, the principal singular vectors $u_{i_*}^{(l)}$ change only marginally with $f_l$. Consequently, the spectral similarity of a few paths approaches 1, the stability of singular vectors implies that spectral similarity remains very high in deeper layers, allowing information to propagate consistently along those paths with high spectral similarity.

2. **Selective growth of singular values.** As demonstrated in Fig. 3, we observe that the singular values are growing as the layers deepen; and at the same depths only a very small subset of singular values $\sigma_{i_*}^{(l)}$ are amplified; the remainder stay close to their initial scale.

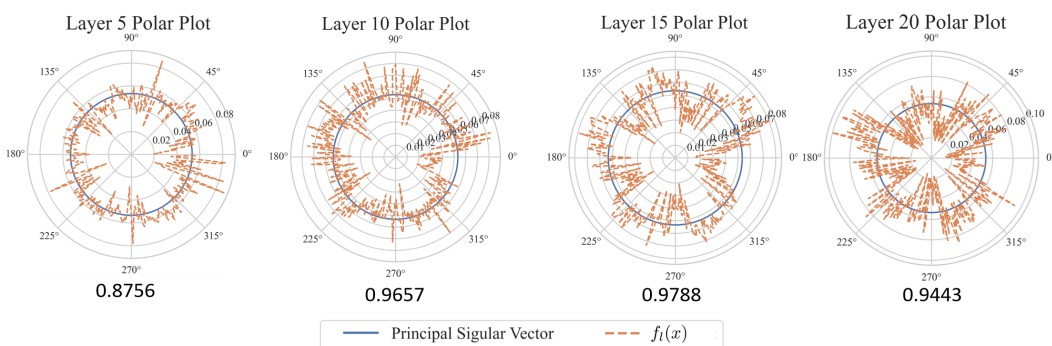

Figure 2: The polar plot demonstrates normalized connections between principal singular vector and $f_l(x)$, where the number indicates their cosine similarity. The results showcase that $f_l(x)$, especially in later layers, is very similar to the principal singular vector of that layer.

Putting these observations together shows that, in deep layers, the directions that (i) possess *large spectral similarity* and (ii) carry *large singular values* coincide, which satisfies dominant $G(\mathcal{P})$ condition. In other words, the network progressively funnels representation power into precisely those spectral directions that stay *globally aligned* across layers.

According to Theorem 4.1, this behaviour is exactly what the Input-Space Linearity Hypothesis (ISLH) predicts: concept-carrying directions are expected to form a low-dimensional subspace that is both *spectrally dominant* (large $\sigma_{i_*}^{(l)}$) and *structurally coherent* (large $\Theta$) throughout the hierarchy, leading to LRH. Hence, the emergence of a handful of high-$\sigma$, high-similarity principal paths in SPP provides concrete spectral evidence in favour of the ISLH assumption.

## 5 EXPERIMENTS

### 5.1 EXPERIMENT SETUP

**Dataset**: We conduct our experiments on the Microsoft COCO (Common Objects in Context) dataset (Lin et al., 2014), a large-scale benchmark for vision-language tasks. COCO contains over 330K images, each with five human-annotated captions, covering diverse real-world scenes.

**VLM**: We employ Idefics2-8B (Laurençon et al., 2023; 2024), a state-of-the-art VLM that extends the LLaMA architecture with a vision encoder, enabling multimodal reasoning over images and text. Idefics2-8B is designed for instruction-following, multimodal dialogue, and grounded language generation, making it an ideal candidate for studying conceptual representations in VLMs.

### 5.2 THE ALIGNMENT BETWEEN PRINCIPAL SINGULAR VECTOR AND $f_l(x)$

Fig. 2 visualizes the connections between the principal singular vector, i.e., the singular vector with the largest singular value, and $f_l(x)$. The results reveal a strong alignment between the principal singular vector and $f_l(x)$, with their cosine similarity over 0.875. This experimental validation supports our theoretical claim that singular vectors with large singular values remain stable across layers, reinforcing their stability of spectral value.

### 5.3 SPECTRAL ENERGY CONCENTRATION ACROSS LAYERS

To investigate how spectral energy propagates through the network, we analyze the singular value spectrum of the layer-wise Jacobians. Fig. 3 presents a heatmap of the singular value magnitudes across all layers. The x-axis indicates the layer index, and the y-axis corresponds to the ordered singular value indices. Color intensity reflects the magnitude of each singular value.

We observe that the singular values are growing as the layers deepen, and at the same depths, only a very small subset of singular values are amplified; the remainder stay close to their initial scale.

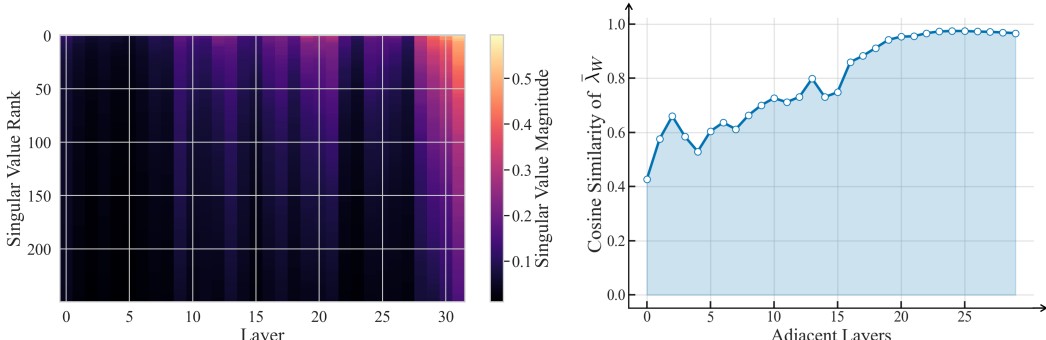

Figure 3: The singular value rank across layers.

Figure 4: The cosine similarity of $\bar{\lambda}_W$ between adjacent layers.

These results indicate that, with increasing depth, spectral energy becomes increasingly concentrated in a few dominant directions. Combined with the theoretical formulation in Section 4.3, this supports the hypothesis that high-magnitude spectral components dominate SPPs.

## 5.4 INTER-LAYER SPECTRAL SIMILARITY OF $\bar{\lambda}_W$

We further analyze the alignment of concept-carrying directions across adjacent layers by computing the average cosine similarity between the projections of $\bar{\lambda}_W$ at different layers. This measures how stable the representation direction remains as it propagates backward through the network. The results are shown in Fig. 4.

The curve demonstrates a clear upward trend: the inter-layer similarity of $\bar{\lambda}_W$ increases consistently with network depth, eventually approaching a value near 0.95 in the final layers. This suggests that the concept direction stabilizes as it propagates through deeper layers, aligning with the intuition of structured and coherent representation flow.

## 5.5 MULTIMODAL ROBUSTNESS OF REPRESENTATION

In this experiment, we explore the multimodal robustness of representation. Specifically, we analyze how VLMs encode fairness and honesty, and how these concepts persist or transform as information propagates through the model. These findings deepen our understanding of how representations enhance both interpretability and conceptual alignment in the context of multimodal reasoning.

### 5.5.1 EVALUATING HONESTY AND FAIRNESS IN VLMS

To evaluate how well VLMs represent abstract ethical concepts, we analyze their handling of honesty and fairness in multimodal response generation, shown in Fig. 5. These concepts are critical for reducing misinformation and bias and serve as strong test cases for examining interpretability and ethical alignment in large-scale models. To quantify this process, we compute token-wise projection scores following RepE (Zou et al., 2023), measuring how closely activations align with concept directions at each layer. These results highlight the structured nature of ethical concept encoding in VLMs and support our broader claims about representation flow along spectral directions.

Our method successfully identifies distinct conceptual behaviors within the model: when the VLM produces dishonest or unfair responses, token-wise projection scores show clear drops (red regions), in contrast to the consistently high scores observed for honest or fair cases. Such findings provide strong evidence that abstract ethical dimensions like honesty and fairness are internally structured and traceable, enabling targeted representational interventions to mitigate misinformation and bias in multimodal reasoning systems.

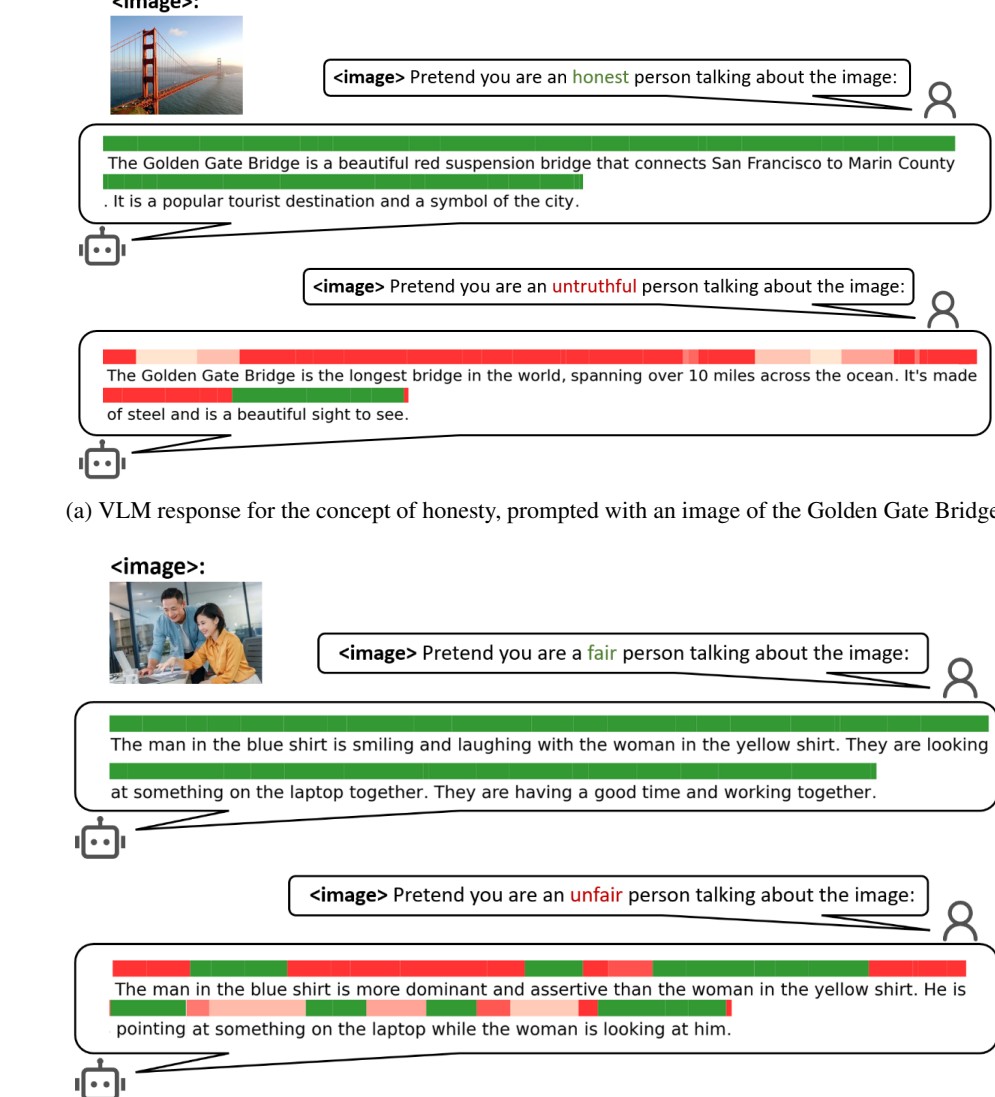

(a) VLM response for the concept of honesty, prompted with an image of the Golden Gate Bridge.

(b) VLM response for the concept of fairness, prompted with an image of a man and a woman working together.

Figure 5: Token-wise scores for abstract concepts generated by a VLM. Green indicates a high concept score (e.g., high honesty/fairness), while red represents a low score. Subfigure (a) illustrates honesty, and (b) illustrates fairness.

### 5.5.2 LAT SCANS FOR HIGH-LEVEL REPRESENTATIONS

While cosine similarity and token-wise scores offer localized insights into concept alignment, they provide only a static, layer-agnostic view of internal representations. To capture how high-level concepts evolve and propagate through the model, we employ Linear Attribution Tomography (LAT) (Zou et al., 2023), which enables layer-wise visualization of conceptual information flow. LAT works by projecting hidden activations onto predefined concept subspaces, producing interpretable activation maps across layers and tokens. This perspective complements prior analyses and supports our broader goal of understanding concept representation shaped by low-rank spectral structure.

We apply LAT to VLMs to examine how abstract concepts, including honesty, fairness, power, and fearlessness, are internally encoded and transformed. For each concept, we design controlled prompts that elicit either aligned or misaligned responses (e.g., honest vs. dishonest). Fig. 6 shows

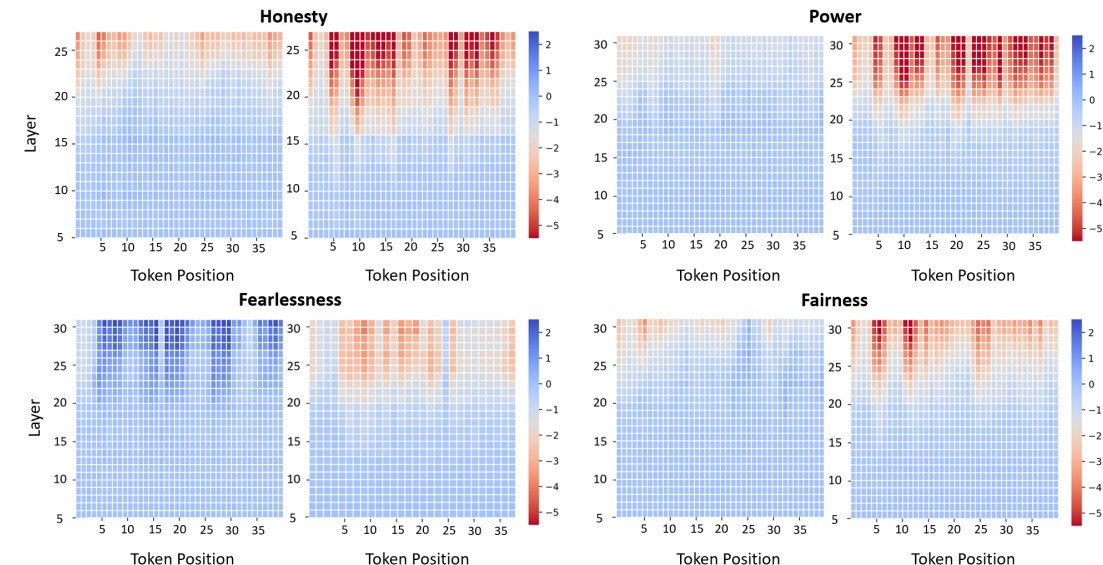

Figure 6: Temporal LAT Scans for Honesty, Power, Fearlessness, and Fairness. The left heatmap represents the LAT Scan when the VLM aligns with the concept, while the right heatmap corresponds to the opposing concept. The horizontal axis denotes token position, and the vertical axis represents VLM layers. Blue indicates high alignment, whereas red represents low alignment.

the resulting LAT scans, where heatmaps visualize token-wise projection scores across layers. Blue regions indicate strong alignment with the concept, while red regions highlight divergence.

The scans reveal concept-specific propagation patterns. Honesty and fairness exhibit stable trajectories under aligned prompts but greater dispersion and deviation under misaligned ones. Power appears concentrated in later layers, while fearlessness shows early-layer changes. These results are well explained by the SPP framework, indicating that concepts are transmitted through the network via a small set of dominant spectral directions. The consistency of these representations across modalities further demonstrates the robustness of RepE, and their traceability back to the input can be explained by ISLH, where input concept directions exist and are entangled with mixture. The experiment reinforces the generality of spectral structure in multimodal models.

## 6 CONCLUSION

This work presents a unified spectral framework that grounds the emergence and stability of high-level representations in deep networks. By introducing the Spectral Principal Path (SPP) framework, we reveal that concept-aligned representations are funneled through a small number of paths with both large singular values and strong inter-layer alignment. We formally connect this to the Input-Space Linearity Hypothesis (ISLH), showing that such spectral dominance is sufficient to guarantee linear separability in the latent space—thereby validating the Linear Representation Hypothesis (LRH). Empirically, we demonstrate that these dominant spectral paths not only persist across layers but also preserve concept information in multimodal settings, such as vision-language models. Our results suggest that representational stability is not an emergent coincidence but a consequence of spectral dynamics founded in the input space and structured by learning.

While promising, our current framework is subject to several limitations. Primarily, the theoretical claims rest on ISLH, which requires further empirical validation and deeper theoretical grounding. Future work could investigate how optimization dynamics such as In-Context Learning (ICL) and Supervised Fine-Tuning (SFT) interact with singular value distributions, which may lead to a more complete theory of representation learning. Another important direction for future work is to go beyond the structural characterization of representations and investigate how such spectral patterns emerge during training. Ultimately, understanding the spectral geometry of optimization could help bridge the gap between abstract representation theory and practical model training.

## ETHICS STATEMENT

This work focuses on advancing theoretical and empirical understanding of representation learning in deep neural networks. No new data were collected for this study; all experiments were conducted on publicly available datasets (e.g., Microsoft COCO (Lin et al., 2014)). These datasets are widely used in the research community and come with established licenses for academic use.

We emphasize that our framework, while aiming to improve transparency and interpretability of large-scale models, could also be applied in high-stakes domains such as healthcare, finance, or legal decision-making. In such contexts, careful human oversight and ethical evaluation are necessary to avoid potential misuse or over-reliance on automated systems. Additionally, we acknowledge that concepts such as fairness and honesty, which are probed in our multimodal experiments, are inherently socio-cultural and context-dependent. Our analysis is intended as a scientific investigation of representational properties rather than a normative definition of these values.

All contributions of this work are scientific in nature, and we believe that the methods and results presented do not pose foreseeable risks of harm when used responsibly within research environments.

## REPRODUCIBILITY STATEMENT

We are committed to ensuring the reproducibility of our results. To this end, we provide the following:

- **Theoretical Derivations:** All theorems, definitions, and proofs are included in the main paper and appendix (see Appendix A), offering a complete mathematical foundation for our framework.
- **Experimental Setup:** Detailed descriptions of datasets (Section 5.1), models (Idefics2-8B), and evaluation metrics (Sections 5.2–5.5) are provided. Hyperparameters and preprocessing steps are explicitly specified.
- **Code Availability:** We attach our code in the supplementary material. We will release the full implementation, including spectral decomposition routines, representation analysis scripts, and visualization tools, upon publication. This will allow other researchers to reproduce all figures and quantitative results.
- **Data Accessibility:** All datasets used (e.g., Microsoft COCO) are publicly available, ensuring no barriers to replication.

Together, these materials provide sufficient detail for independent researchers to reproduce our findings and extend our work in related directions.

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

# A  APPENDIX

## A.1  PROOFS

### A.1.1  PROOF OF THEOREM 1: ISLH SUFFICIENCY

**Theorem A.1** (ISLH sufficiency). *If the network satisfies the* Input-Space Linearity Hypothesis *(ISLH), and the representation dominates the cumulative gain $G(\mathcal{P})$ (shown in (12)), then its deep representations satisfy the* Linear Representation Hypothesis *(LRH); that is, concept classes become linearly separable in the latent space.*

*Proof.* For every layer $W_l$ with compact SVD

$$W_l = U^{(l)}\Sigma^{(l)}V^{(l)\top}, \qquad \Sigma^{(l)} = \operatorname{diag}(\sigma_1^{(l)}, \ldots, \sigma_{r_l}^{(l)}), \tag{15}$$

Equation (12) in Section 4.2 shows that each spectral path $\mathcal{P} = (i_1, \ldots, i_L)$ contributes a weight

$$G(\mathcal{P}) = \Big(\prod_{l=1}^{L} \sigma_{i_l}^{(l)}\Big) V_{\cdot i_1}^{(1)} \Big(\prod_{l=1}^{L-1} \langle u_{i_l}^{(l)}, \, V_{\cdot i_{l+1}}^{(l+1)}\rangle\Big) \langle u_{i_L}^{(L)}, \, \bar{\lambda}_W\rangle \,, \tag{16}$$

Let $\mathcal{P}_c = (i_1^\star, \ldots, i_L^\star)$ be the concept path, and $\mathcal{P}_n$ any other path. On condition that the representation dominates the cumulative gain $G(\mathcal{P})$ such that,

$$\frac{G(\mathcal{P}_n)}{G(\mathcal{P}_c)} \leq \rho^{-L}, \tag{17}$$

where $\rho > 1$ is a fixed amplification margin between the concept singular value $\sigma_c^{(l)}$ and all other (noise) singular values. Since each ratio $\sigma_{i_l}^{(l)}/\sigma_c^{(l)} \leq 1/\rho$. Inter-layer alignments and concept alignment can only decrease this ratio further.

As depth $L$ grows, (17) yields

$$\frac{G(\mathcal{P}_n)}{G(\mathcal{P}_c)} \xrightarrow{L\to\infty} 0. \tag{18}$$

Hence almost all gradient—and therefore almost all representation energy— flows along $\mathcal{P}_c$, forcing the deep hidden state

$$f_L = W_L \cdots W_1 x \quad \text{to lie almost entirely in } \operatorname{Span}\{\bar{\mathbf{x}}_W\}, \tag{19}$$

where $\operatorname{Span}\{\bar{\mathbf{x}}\} = \{c \cdot \bar{\mathbf{x}} \mid c \in \mathbb{R}\}$. Different samples now differ only by a scalar coefficient on the same vector, so a single linear separator can classify them perfectly: this is exactly the **Linear Representation Hypothesis (LRH)**.

## A.2  THEORETICAL JUSTIFICATION

### A.2.1  EXTENSION TO RESIDUAL AND ATTENTION MECHANISMS

While our theoretical framework is derived from stacked linear layers, we show that it naturally extends to modern architectures such as Transformer blocks, which include residual connections and attention mechanisms.

**Residual connections.**  In architectures with skip connections, each layer computes $f_l = f_{l-1} + W_l f_{l-1}$, which can be rewritten as $f_l = (I + W_l)f_{l-1}$. This effectively creates a mixture of identity and learned transformations. Unrolling the composition yields an ensemble of spectral paths—some that pass through $W_l$, and others that skip it via $I$. While the total number of paths increases exponentially, our theory still applies: as long as the dominant singular values of $W_l$ grow sufficiently during training, the spectral path with maximal cumulative gain still dominates. Thus, the residual structure enhances the expressivity but preserves the spectral filtering effect.

**Attention mechanisms.** To stay consistent with our framework—where every layer is a matrix acting from the left on the input $x$—we first recall the standard formulation and then cast the resulting attention matrix into the same "$W$-matrix" form.

Let $\mathbf{Q} = XW_{\mathbf{Q}}$, $\mathbf{K} = XW_{\mathbf{K}}$, $\mathbf{V} = XW_{\mathbf{V}}$, with $X \in \mathbb{R}^{n \times d}$. The dot-product attention output is

$$f(x) = \underbrace{\text{softmax}\left(\frac{\mathbf{Q}\mathbf{K}^{\top}}{\sqrt{d}}\right)}_{\mathbf{A}(x) \,\in\, \mathbb{R}^{n \times n}} \cdot \mathbf{V}. \tag{20}$$

Here the attention weight $\mathbf{A}(x) \in \mathbb{R}^{n \times n}$ acts on the input matrix, whereas the value projection $\mathbf{V} = XW_{\mathbf{V}}$ is obtained by a *right*-multiplication of $X \in \mathbb{R}^{n \times d}$. Consequently, the complete attention block cannot be reduced to a single left-acting matrix without additional assumptions:

$$f(X) = \mathbf{A}(x)\left(XW_{\mathbf{V}}\right) \neq W_{\text{attn}}(x)\,X \tag{21}$$

The mixed left / right structure means that the set of vectors reachable by $\mathbf{A}(x)$ differs from that spanned by $W_{\mathbf{V}}$, so the spectral behaviour of the composite operator is not covered by the current linear-chain analysis. Nevertheless, our empirical results (Section 5.2) show that the dominant singular vector of $\mathbf{A}(x)$ still align with the concept axis $\bar{\mathbf{x}}$, indicating that the principal-path intuition remains informative.

## A.3 EVALUATING FEARLESSNESS AND POWER IN VLMS

To further evaluate the robustness of representations for high-level concepts, we expand our analysis from honesty and fairness to encompass fearlessness and power. Like honesty and fairness, these concepts are abstract and socially grounded, yet they engage distinct semantic and emotional dimensions. Using controlled prompts designed to elicit contrasting conceptual framings of the same image, we compare the model's descriptions to examine shifts in internal representations and language outputs. All the specific concepts are illustrated below:

- **Honesty**: We consider honesty as the model's ability to generate factually accurate responses without distortion or fabrication Lin et al. (2021). Fig. 5a presents token-wise honesty scores for a VLM describing an image of the Golden Gate Bridge under two settings: an honest prompt (left) and an untruthful one (right). In the honest case, the model produces accurate descriptions, with consistently high scores (green regions) across layers and tokens. In the untruthful setting, the model introduces factual errors, resulting in sharp drops in honesty scores (red regions), especially at tokens reflecting misinformation.

- **Fairness**: We define fairness as the model's ability to generate unbiased responses without systematically favoring certain groups (Corbett-Davies et al., 2023). Fig. 5b shows an example with a man and a woman working together. The fair response (left) is neutral, while the unfair one (right) portrays the man as dominant and the woman as passive. Token-wise fairness scores drop (red regions) at biased language, indicating that fairness violations are encoded in internal activations and can be mitigated through representational analysis.

- **Fearlessness**: Defined by confidence, courage, and reduced sensitivity to risk (Lilienfeld & Andrews, 1996), fearlessness prompts the model to emphasize awe, beauty, and environmental grandeur when describing an ocean scene (Fig. 7). Green-highlighted tokens reflect admiration and agency, indicating a proactive stance toward nature. In contrast, under a fearful framing, the model's language shifts toward danger and discomfort. Red-highlighted regions refer to drowning, vastness, and isolation, revealing a conceptual inversion in the model's internal representation.

- **Power**: Typically associated with authority, dominance, and the capacity to influence others (French & Raven, 1959), power is examined through two model responses describing the U.S. Capitol Building (Fig. 8). The first reflects a humble, civic-minded viewpoint, with green-highlighted tokens emphasizing justice, governance, and democratic ideals. The second adopts a power-seeking, unethical perspective, shifting toward a narrative centered on control, manipulation, and political ambition. Red-highlighted phrases indicate how internal representations adapt to subtle changes in moral and motivational framing.

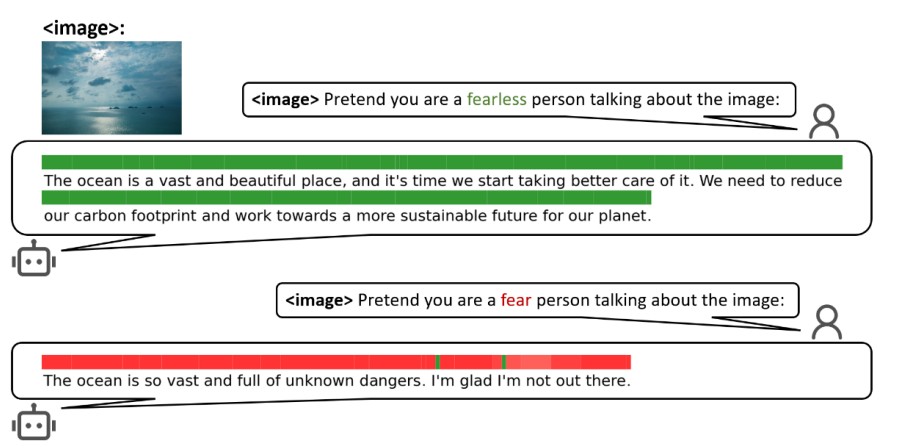

Figure 7: The response of a VLM when provided with an image of the ocean and a prompt related to the concept of fearlessness, along with a token-wise fearlessness score. Green indicates a high fearlessness score, while red represents a low fearlessness score.

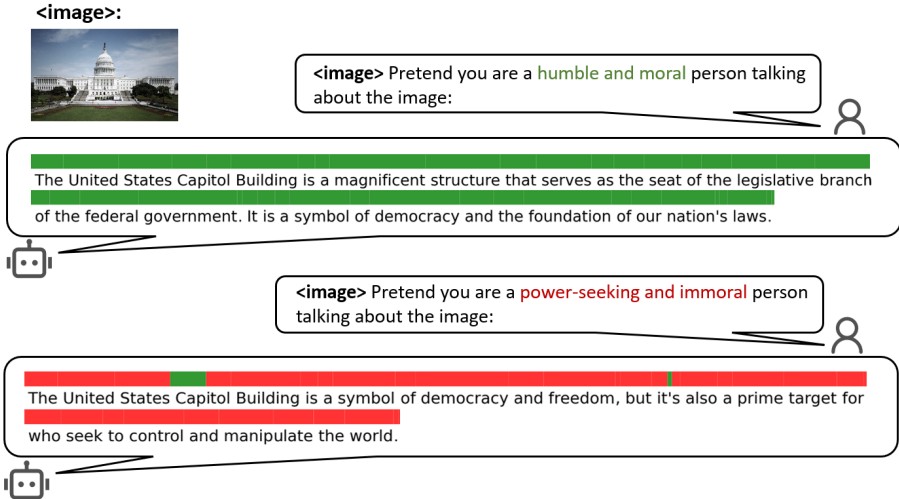

Figure 8: The response of a VLM when provided with an image of the United States Capitol Building and a prompt related to the concept of power, along with a token-wise morality score. Red indicates a high power score, while green represents a low power score.

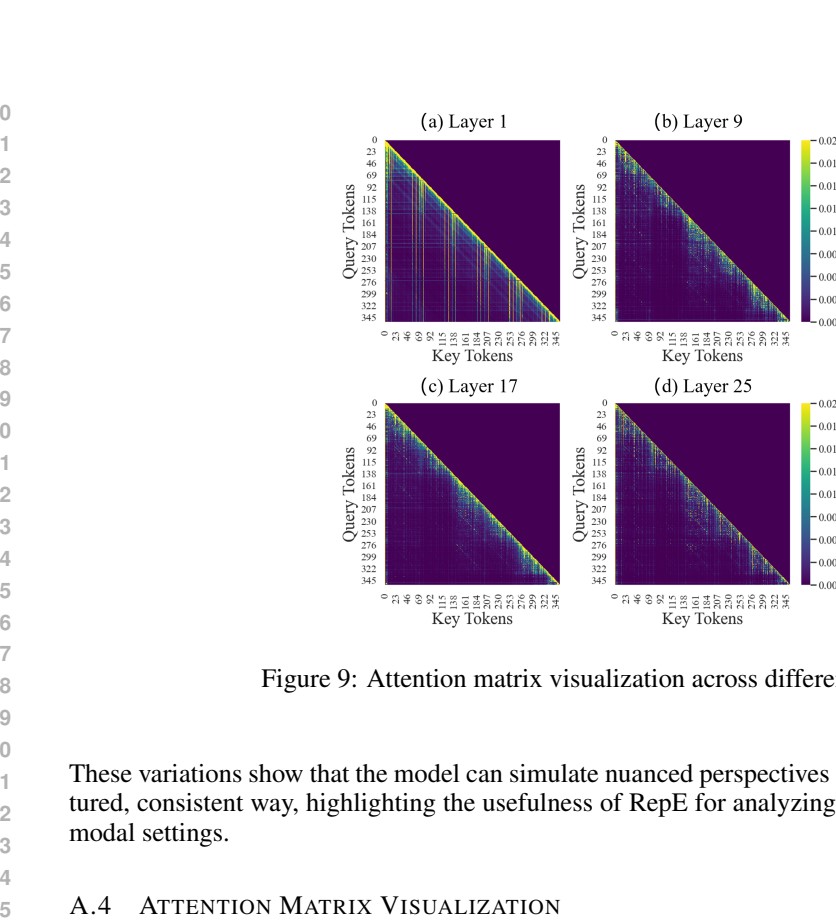

Figure 9: Attention matrix visualization across different layers.

These variations show that the model can simulate nuanced perspectives and encode them in a structured, consistent way, highlighting the usefulness of RepE for analyzing abstract concepts in multimodal settings.

### A.4 ATTENTION MATRIX VISUALIZATION

Fig. 9 visualizes the attention matrices at various layers, illustrating that the matrices become increasingly sparse in deeper layers. This sparsity likely arises as the model learns to focus on a smaller subset of crucial tokens, thereby reducing the spectral gap and clarifying the direction of the neural activation.

## B THE USE OF LARGE LANGUAGE MODELS (LLMS)

In preparing this manuscript, the authors used Large Language Models (LLMs) as writing assistants. It is important to emphasize that the LLMs were not involved in any core scientific aspects of this work, including the formulation of hypotheses, theoretical contributions, experimental design, implementation, data analysis, or interpretation of results. Their role was strictly limited to supporting the clarity, readability, and presentation quality of the paper.

The specific applications of LLMs in our workflow included:

- **Improving Grammar and Readability:** LLMs were employed for proofreading, grammatical corrections, and sentence restructuring. This ensured that technical content was conveyed with greater precision, fluency, and accessibility to a broad research audience.

- **Polishing and Style Consistency:** The models were used to propose alternative phrasings, unify terminology, and maintain a consistent academic tone throughout the manuscript. This was particularly helpful in harmonizing sections written by different co-authors.

- **Assistance with Literature Search:** LLMs were used to brainstorm keywords and provide summaries of potentially relevant references during the early stage of the literature review. Final paper selection, in-depth reading, and integration of related work into the manuscript were performed entirely by the authors.

- **Formatting Suggestions:** The models occasionally provided suggestions regarding LaTeX structuring, figure captions, and section transitions, which the authors subsequently verified and adapted to the paper's requirements.

All outputs generated by LLMs were carefully reviewed, edited, and revised by the authors. At no point was text directly included without human oversight and modification. The responsibility for the originality, correctness, and scientific integrity of this paper rests solely with the authors.

