# OpenReview forum: "Structure before the Machine: Input Space is the Prerequisite for Concepts"
_ICLR.cc/2026/Conference — Submitted to ICLR 2026_

### Official Review · Reviewer_SVti · 2025-10-27

**Soundness:** 2
**Presentation:** 2
**Contribution:** 2
**Rating:** 2
**Confidence:** 3

**Summary:**

This paper proposes Input-Space Linearity Hypothesis (ISLH), extending the Linear Representation Hypothesis (LRH) to the input space where each sample is treated as an entangled mixture. The paper then proposes a Spectral Principal Path (SPP) framework, which provides an explanation for LRH based on ISLH, that there exists spectral paths along certain singular directions to amplify and propagate the linear direction from inputs to outputs. It then runs experiments from a reference paper, RepE, on VLMs, and obtains similar results to the reference.

**Strengths:**

The spectral path viewpoint is new and intriguing.

The paper writes clearly and is easy to read.

**Weaknesses:**

Unclear or missing description of experimental setup. For example, what is the data used to plot Figures 2, 3, and 4? What is x here? What is the concept used (W) to obtain Figure 4? Further, how is the conclusion “only a very small subset of singular values are amplified; the remainder stay close to their initial scale” obtained from Figure 3?

Disconnection between the framework and experiments. (1) The framework builds upon inputs in a vector space propagated through a generalized network, while the experiments are launched on Idefics2-8B, which consists of two encoders for text and image modalities, respectively. Though these two modalities converge into a single embedding space, it is unclear whether the framework is useful for visual inputs, since later (contracting concept) experiments are all based on text inputs. Still, for text inputs, it is unclear whether they fulfill the basic assumption that “each sample is an entangled mixture” in a vector space. (2) Section 5.5 is detached from the framework. According to the description, they are literally a replication of what has been done in [1], on VLM instead of LLM. The plots do not provide sufficient support to the proposed framework or help claim its validity.

[1] Representation Engineering: A Top-Down Approach to AI Transparency

**Questions:**

See weaknesses. Besides the above points:

Why do you call Section 5.5.1 (line 373) your method? What did you modify compared to the original algorithm in [1], used to plot Figure 9 in their Section 4.3.2?

Also, which layer in which module (vision or text) did you choose? Notice in [1], the authors clearly stated that they did a sweep to identify one of the strongest layers for concept reading and picked that one. This is related to your description in Line 424 (“layer-agnostic”), which is incorrect.

Misc: (typos) in line 148 concept; line 237 singular.

[1] Representation Engineering: A Top-Down Approach to AI Transparency

---

### Official Review · Reviewer_XQ8K · 2025-10-31

**Soundness:** 3
**Presentation:** 3
**Contribution:** 3
**Rating:** 4
**Confidence:** 4

**Summary:**

This paper introduces the Spectral Principal Path (SPP) framework to explain the emergence and stability of linear representations in deep networks. The authors propose the Input-Space Linearity Hypothesis (ISLH)—an extension of the Linear Representation Hypothesis (LRH)—which suggests that concept-aligned directions originate in the input space and are selectively amplified through network depth.
The authors conside a derivation based on the Jacobian SVD decomposition paired with cumulative gain maximization, arguing that representations found in the input space propagate along a few dominant spectral paths aligned with the largest singular values.

The experiments show that principal singular vectors stabilize in deeper layers, singular values exhibit selective growth, and concept directions become increasingly concentrated and stable, supporting the theoretical claims.

**Strengths:**

The work bridges theory and interpretability, offering an appealing spectral perspective on how neural networks encode and stabilize concepts.

The proposed Spectral Principal Path framework provides a unified spectral mechanism that connects the input-space structure to the linear separability observed in deep representations, while this link is well articulated in the main text.

Moreover, the provided evidence of singular vector stabilization and selective singular value growth provide some important insights into the observed representational coherence.

**Weaknesses:**

My main concerns revolve around some experimental choices of the authors and some restricting assumptions considered. Specifically:

*Simplified architecture assumption*: The SPP derivation assumes a purely stacked linear model, which may not capture non-linearities or normalization effects critical in deep networks. How robust is the theory under more realistic settings? Would the insights provided in the manuscript extend to more common architectures?

*Residual and attention extensions*: In the main text, line 214, the authors mention "we show our extension to residual connections and attention mechanisms". I find this to be a bit misleading; while residual connections are somewhat discussed theoretically (on a single paragraph), the attention mechanism is only empirically justified. Could the authors provide a more formal link between the residual connections/attention operators and the spectral path analysis? If not, is the only way to validate if the findings hold through a questionable in terms of interpretation empirical evaluation?

*Limited model diversity*: The experiments rely on a single model (Idefics2-8B). How general are the observed SPP behaviors across other architectures (e.g., LLMs, CNNs, or smaller-scale models)?

*Evaluation Setup*: Testing SPP behavior in binary “concept flip” or contrastive settings (inspired by RePe) seems restrictive. Would the same spectral alignment hold in continuous or open-ended concept spaces?

*Interpretation of multimodal results*: While the LAT scans are visually compelling, it’s unclear how they directly confirm spectral path propagation rather than post-hoc representational clustering. Similar results where provided in the original LAT paper, that does not consider SPP.

**Questions:**

Please see the Weaknesses section.

---

### Official Review · Reviewer_MB12 · 2025-11-01

**Soundness:** 1
**Presentation:** 1
**Contribution:** 2
**Rating:** 2
**Confidence:** 4

**Summary:**

The paper claims that concept directions already exist in the input space, and  neural networks propagate these directions along a few dominant paths layer by layer. Assuming this structure (ISLH), the authors argue that the LRH naturally emerges. The paper expands layer-wise Jacobians via SVD and defines a "path gain" as the product of singular values and inter-layer alignments. A few experiments are conducted on a VLM to probe intermediate-layer concept alignment.

**Strengths:**

1. Analysis of the LRH and the inquiry for the reason of its emergence is interesting, analysing it through intermediate layers is sound.

**Weaknesses:**

1. LRH has mainly been discussed in text-only contexts, and the ISLH is also formulated that way. Yet the experiments use a VLM, and all actual manipulations seem to be on the language side. It’s unclear what the vision modality contributes here, this effectively invalidates the claimed "raw-space" perspective.
2. The exposition is confusing (see Questions).
3. Theoretical claims are made for linear networks. Since a purely linear network can be collapsed into a single matrix, it’s unclear how this is informative for real nonlinear architectures.
4. Section 5.2 largely repeats Section 4.3 without adding new content.
5. Experimental evaluation is weak and mostly qualitative.
6. 5.5.1 repeats contents of 5.5.2.

**Questions:**

1. Eq 12: is G(\mathcal{P}) appears to not be a scalar that can be maximized - can the authors clarify?
2. How to interpret Fig 2: what do the angles represent?

---

### Official Review · Reviewer_UAP6 · 2025-11-03

**Soundness:** 2
**Presentation:** 3
**Contribution:** 3
**Rating:** 4
**Confidence:** 3

**Summary:**

The paper develops a theoretical framework aimed at understanding how linear representations of high-level concepts emerge in deep neural networks. The authors hypothesize that observations in the ambient space are a linear combination of the high-level concept and spurious linear directions. This is coined the Input Space Linearity Hypothesis (ISLH).

Building on this, the authors derive the Spectral Principal Path (SPP) for an affine network without non-linearities. The SPP is the path through projection matrices for which output-input singular vectors of adjacent matrices remain mostly co-linear, whilst corresponding singular values are large. Through Theorem 4.1, the ILHR and SPP framework is linked to the Linear Representation Hypothesis, connecting the work with prior research.

**Strengths:**

The paper is well-written, quite self-contained, and has well-crafted illustrations.

The proposed method is both creative and original. The authors honestly point out that “our current framework is subject to several limitations.” (conclusion).

While the developed method rests on a few key assumptions, it is elegant and well thought through. Given that the assumptions hold in practice, which the presented results suggest, the framework is a notable step towards better understanding how linear representations of high-level concepts emerge and propagate through deep neural networks.

**Weaknesses:**

1. Introduction: covers the core concepts; however, readers unfamiliar with the LRH paper (Park et al. 2023) have considerably less context for understanding the setting.

1. The theoretical derivation of the SPP method hinges on having a generalized network, like equation 5, without non-linearities. Yet, little discussion is provided on the limitations that come with this assumption.

1. It wasn’t immediately clear to me why the second term in equation 8 is inserted. Based on the model specification in equation 5, this term would be zero. It would be good to state more explicitly for which case(s) this term is non-zero, e.g., referring to Appendix A.2.1.

1. While the authors state in their reproducibility statement that they are committed to releasing the code, no private repository was provided as part of the review. This complicates disambiguating parts of the work, specifically related to these points:

    1. Complexity of computing the SPP (equation 13): Finding the principal path is exponential in the depth of the model. Hence, for a $L$ layer network where the layer-wise jacobians have ranks r_1, …, r_L, the possible paths are: $\prod_{l=1}^L r_l$. While this could become prohibitive for deep networks, it is not something that the authors discuss nor provide details on how to compute in practice.
    1. It is not immediately clear how the results in Figures 2 and 4 were computed. Please see the questions below for more details on where the confusion lies. Without further elaboration, it raises questions about the robustness and generality of the proposed framework.

    1. In line with the above, I am left with unanswered questions in terms of how rigorously the proposed method has been evaluated. At the outset, the results seem mostly qualitative, which weakens the support for ISLH and SPP.

**Questions:**

1. Line 51: Maybe it would be useful to introduce the notion of “unembedding space”, this concept is not really a “standard” standard concept.

1. Line 139: “recent advances in interpretability have shifted the focus toward analyzing representation”, please give references to this statement.

1. Line 148: “co ncept” -> “concept”

1. Eq. 3+4, should $x$ not be $\bf{x}$?

1. Figure 2: It is unclear how the cosine similarities in Figure 2 were computed; was a single sample $x$ used, or were cosine similarities between principal singular vector(s) and multiple samples for the same and/or distinct concepts used?

1.  Figure 2: How should the polar plot be interpreted exactly? Is it the components of $f_l(x)$ that are plotted as the dashed lines, or how exactly was this plot constructed?

1. Figure 3: You write in the context of the figure (Line 323), “only a very small subset of singular values are amplified”. This interpretation seems a bit forced, considering that several layers have many non-zero magnitude singular values. Also, the relative difference between the highest singular value remains quite small compared to other non-zero singular values e.g., layers 19-21. Could you elaborate on Line 323 in relation to Figure 3?

1. Figure 4: How did you compute the concept direction $\bar{\lambda}_W$ for the results? Was a single concept used for this computation, or are the cosine similarities across layers an average for multiple concept directions?

1. Figure 5: What constitutes a “low” and a “high score?

1. Section 5.5: Experiments in sec 5.5, this is quite qualitative, can this not be made more quantitative?

1. Conclusion: Should this be written in past tense?

---

### Meta-Review · Area_Chair_4JKy · 2026-01-05

**Summary:**

This work studies the formation of linear representations in neural networks, and it looks quite interesting. However, due to the low review score and the lack of response from the authors, I deem that the authors have given up, and I thus recommend rejection.

**Reviewer Scores:**

NA

---

### Decision · Program_Chairs · 2026-01-26

Reject